There are amendments to this paper

# Enhancement of and interference among higher order multipole transitions in molecules near a plasmonic nanoantenna

Evgenia Rusak[1], Jakob Straubel[1], Piotr Gładysz[2], Mirko Göddel[1], Andrzej Kędziorski[2], Michael Kühn [3], Florian Weigend[4], Carsten Rockstuhl [1,4] & Karolina Słowik [2]*

Spontaneous emission of quantum emitters can be modified by their optical environment, such as a resonant nanoantenna. This impact is usually evaluated under assumption that each molecular transition is dominated only by one multipolar channel, commonly the electric dipole. In this article, we go beyond the electric dipole approximation and take light-matter coupling through higher-order multipoles into account. We investigate a strong enhancement of the magnetic dipole and electric quadrupole emission channels of a molecule adjacent to a plasmonic nanoantenna. Additionally, we introduce a framework to study interference effects between various transition channels in molecules by rigorous quantum-chemical calculations of their multipolar moments and a consecutive investigation of the transition rate upon coupling to a nanoantenna. We predict interference effects between these transition channels, which allow in principle for a full suppression of radiation by exploiting destructive interference, waiving limitations imposed on the emitter's coherence time by spontaneous emission.

[1] Institute of Theoretical Solid State Physics, Karlsruhe Institute of Technology, 76131 Karlsruhe, Germany. [2] Institute of Physics, Faculty of Physics, Astronomy and Informatics, Nicolaus Copernicus University, Grudziadzka 5, 87-100 Torun, Poland. [3] Institute of Physical Chemistry, Karlsruhe Institute of Technology, 76131 Karlsruhe, Germany. [4] Institute of Nanotechnology, Karlsruhe Institute of Technology, 76021 Karlsruhe, Germany. *email: karolina@fizyka.umk.pl

In recent years, optical nanoantennas have been suggested to modify the radiative properties of point sources[1]. Optical nanoantennas, with sizes in the order of hundreds of nanometers, are commonly defined as devices designed to efficiently convert free-propagating optical radiation to localized energy, and vice versa. In plasmonics, metallic nanoantennas are used to confine electromagnetic fields to volumes smaller than the diffraction limit. This happens as the conduction electrons of the plasmonic nanoantenna can be driven by the incident electric field in a resonant collective oscillation known as a surface plasmon polariton.

The huge field enhancement and related spatial energy confinement leads to a boost of the energy exchange rate between a quantum emitter and the electromagnetic field. Resulting shortened interaction timescales are the reason for a suppression of spontaneous emission lifetimes through the Purcell effect by multiple orders of magnitude[2]. For quantum emitters located only a few nanometers from the nanoantenna surface a quenching effect was reported[3]. However, it does not imply suppressed lifetimes: an important reason for quenching is that the spontaneously emitted photons are rather absorbed by the nanoantenna than radiated into the far field[4]. Still, the transition occurs at a Purcell-enhanced rate. Here we add an additional layer of complexity to the control of spontaneous emission rates, i.e., either their enhancement or suppression, through tailored interference of different light-matter interaction channels.

Usually, only the electric-dipole contribution to a quantum-mechanical state transition of an emitter is considered. This is often justified by the negligible spatial variation of the electric field over the size of the emitter[5], although recently studies have been made of spatially extended emitters adjacent to plasmonic nanoantennas[6]. In presence of a nanoantenna, the electric field is localized into nanometric spatial domains, providing high intensities and spatial modulations at the length scale of a molecule[7]. Thus, higher-order multipolar contributions to light-matter coupling may become relevant. Until now, the enhancement of magnetic-dipole emission was studied near metallic[8-10] or dielectric[11,12] nanostructures or within focused laser beams[13]. Large enhancements of electric-quadrupole transitions by nanostructures were also predicted[14-16]. Transitions driven with several multipolar mechanisms were observed in semiconductor quantum dots[17].

In these works, the considered transitions were usually assumed to be either purely electric or magnetic dipolar, or electric quadrupolar. However, depending on their symmetry properties dictated by their geometry, quantum emitters can have transitions with contributions from different multipoles at once. Each of these multipoles can be enhanced near a nanostructure, as predicted in ref. [18] and demonstrated experimentally in ref. [19]. If these contributions correspond to a transition between the same pair of eigenstates, interference effects are expected[20]. In ref. [17], the ability of semiconductor quantum dots to probe electric and magnetic fields simultaneously was shown, stating interference between higher-order decay channels of the quantum dots.

In this work, we study the interference effects between different multipole transition channels by coupling a molecule to a plasmonic nanoantenna in a conceptually and operationally simple approach based on Fermi's golden rule. We consider a patch nanoantenna of a geometry rich enough to allow for different effects to be discussed, from selective enhancement of different multipolar contributions, to constructive or destructive interference thereof. We demonstrate the tunability of such nanoantenna with respect to magnetic-dipole and electric-quadrupole enhancement in a range of wavelengths from mid-optical to near-infrared. Next, the geometry is tuned to a specific transition of a

specific molecule, characterized by a set of multipolar moments, that have been calculated with quantum-chemical methods. Depending on the molecular orientation with respect to the nanoantenna, transition rate enhancement in various channels is demonstrated. Consequently, we show that in deliberately chosen systems, i.e., combinations of molecules and optical nanoantennas, it must be possible to suppress the emission comparable to what has been suggested long time ago while using spatially extended photonic crystal structures with a well defined optical band gap[21]. The possibility of suppression is especially appealing since it implies correspondingly longer spontaneous emission lifetimes. This largely increased lifetime of the emitter is crucial for applications such as quantum information processing and storage, where a short lifetime of the emitter is a serious constraint. This constraint might be relaxed in schemes based on our work.

## Results

**Different multipolar contributions to transition rates**. Our approach is based on Fermi's golden rule, which accounts for the molecular transition rate depending on the nanoantenna-scattered electromagnetic field properties and on molecular characteristics. A theoretical prediction for the transition rate $\Gamma$ from an initial state $|i\rangle$ with energy $\hbar\omega_i$ to a final state $|f\rangle$ of energy $\hbar\omega_f$ is given by[22]

$$\Gamma = \frac{2\pi}{\hbar^2}|\langle f|V|i\rangle|^2 \rho(\omega_i - \omega_f), \qquad (1)$$

with the reduced Planck constant $\hbar$. The density of states $\rho(\omega)$ needs to be taken at the quantum emitters transition frequency $\omega_i - \omega_f$. The interaction Hamiltonian $V$ is studied up to the electric-quadrupolar order[23]

$$V = -\underbrace{\mathbf{p} \cdot \mathbf{E}(\mathbf{r}_0)}_{V_{ED}} - \underbrace{\mathbf{m} \cdot \mathbf{B}(\mathbf{r}_0)}_{V_{MD}} - \underbrace{[\mathbf{Q}\nabla] \cdot \mathbf{E}(\mathbf{r}_0)}_{V_{EQ}}, \qquad (2)$$

where $\mathbf{r}_0$ is the location of the molecule, and the electric-dipole ED, magnetic-dipole MD and electric-quadrupole EQ contributions are included (Please note that in the above Hamiltonian the elements of the quadrupole moment operator are defined as $Q_{kl} = \frac{e}{2}r_k r_l$, but they can be replaced by the traceless form $Q_{kl} = \frac{e}{2}(r_k r_l - \frac{1}{3}\delta_{kl}r^2)$ in source-free regions, where $\nabla \cdot \mathbf{D} = 0$.). The Hamiltonian in Eq. (2) includes the electric $\mathbf{E}$ and magnetic $\mathbf{B}$ fields induced by a combination of considered multipolar sources, scattered by the nanoantenna and evaluated at the molecular position. The fields are calculated classically (see Methods and Supplementary Note 2). The transition moments in Eq. (2), i.e., the matrix elements of the electric dipole $\mathbf{p}$, magnetic dipole $\mathbf{m}$, and electric quadrupole $\mathbf{Q}$, are calculated between the initial and final states of the molecule with time-dependent density functional theory (TDDFT), as described in Methods. The square bracket in Eq. (2) contains a matrix multiplication of the electric-quadrupole moment tensor and the column vector $\nabla = (\frac{\partial}{\partial x}, \frac{\partial}{\partial y}, \frac{\partial}{\partial z})^T$, while $T$ denotes operation of transposition.

To describe spontaneous emission, the interaction Hamiltonian (2) should account for single-molecule-to-single-photon coupling[5], i.e., the electromagnetic fields around the nanoantenna should be normalized to values corresponding to single-photon excitations. This assures that the resulting transition rate $\Gamma$ scales with the square of transition moments as expected. Proper normalization factor can be based on free-space emission rates for different multipoles and the corresponding Purcell enhancement factors $F_{tot}^{MO,\varphi}(\mathbf{r}_0, \omega) = P_{na}^{MO,\varphi}(\mathbf{r}_0, \omega)/P_0^{MO}(\omega)$. Purcell factors are classically calculated for each type of emitter, by evaluation of powers of the emitter in the presence of the nanoantenna

$P_{na}^{MO,\varphi}(\mathbf{r}_0, \omega)$ and in free space $P_0^{MO}(\omega)$. Here the multipolar operator $MO \in \{ED, MD, EQ\}$ denotes the type of emitter, and $\varphi$ indicates that the result depends on the orientation of the emitter with respect to the nanoantenna. Identification of the emitted powers rescaled by single-photon energy $P_{na}^{MO,\varphi}/\hbar\omega$ $(P_0^{MO}/\hbar\omega)$ with the transition rates $\Gamma^{MO,\varphi}$ $(\Gamma_0^{MO})$ of a quantum emitter (e.g., a molecule) allows us to write

$$\underbrace{\frac{2\pi}{\hbar^2}|\langle f|V_{ED}|i\rangle|^2\rho}_{\Gamma^{ED,\varphi}} = F_{tot}^{ED,\varphi}\underbrace{\frac{\omega^3|\mathbf{p}|^2}{3\pi\epsilon_0\hbar c^3}}_{\Gamma_0^{ED}}, \qquad (3)$$

where we have used Eq. (1) and the definition of the total Purcell enhancement factor as a ratio of powers extracted from an oscillating dipole or quadrupole in the presence of a photonic nanostructure and in free space. Above, $\epsilon_0$ stands for the vacuum electric permittivity, and the expression for $\Gamma_0^{ED}$ on the right-hand side is given by the Weisskopf-Wigner formula[5]. Similarly, for magnetic-dipole and electric-quadrupole sources we have

$$\underbrace{\frac{2\pi}{\hbar^2}|\langle f|V_{MD}|i\rangle|^2\rho}_{\Gamma^{MD,\varphi}} = F_{tot}^{MD,\varphi}\underbrace{\frac{\omega^3|\mathbf{m}|^2}{3\pi\epsilon_0\hbar c^5}}_{\Gamma_0^{MD}}, \qquad (4)$$

$$\underbrace{\frac{2\pi}{\hbar^2}|\langle f|V_{EQ}|i\rangle|^2\rho}_{\Gamma^{EQ,\varphi}} = F_{tot}^{EQ,\varphi}\underbrace{\frac{\omega^5\sum_{i,j}|\mathbf{Q}_{ij}|^2}{10\pi\epsilon_0\hbar c^5}}_{\Gamma_0^{EQ}}, \qquad (5)$$

where we have used the free-space MD and EQ transition rates[20]. Based on this equality we can jointly normalize fields in $V_{MO}$ and density of electromagnetic states $\rho$. Since any kind of multipole can in general be a source of both electric and magnetic fields, to be scattered by the nanoantenna, we normalize each field from a given source by the same factor, preserving phase relations. Obviously, with the proposed scheme we retrieve expected free-space transition rates. Based on these normalized fields, we can also consider subsequently an arbitrary superposition thereof that requires a weighting of each term according to the multipolar moments of the transitions of a specific molecule. Please note, when reconstructing the fields that are considered in Eq. (2), the contribution of each multipolar emitter to the electric and magnetic field and the electric field gradient is taken into account. We use the total field, i.e., the one emitted by the actual multipolar source and the secondary field scattered by the antenna. The procedure to extract the fields at the point of interest is described in the Supplementary Note 2.

We now make a comment on quantifying influence of the photonic environment, i.e., the nanoantenna. Canonically, the influence of nanoantennas or cavities in general on spontaneous emission of quantum systems is expressed in terms of modified density of states $\rho(\omega)$[24]. It accounts for field enhancement at given source's location and spectral dependencies. We stress that in the method described in this work, in contrast to the usual approach, the same influence is taken into account through the modification of field profiles $\mathbf{E}(\mathbf{r})$ and $\mathbf{B}(\mathbf{r})$: it is the field that is enhanced and reshaped, rather than density of states. This allows us to assume that the density of states is common for all transition mechanisms. Naturally, it makes only sense to consider the impact of the photonic environment in only one of the two terms, $V$ or $\rho$, that appear in Eq. (1). Here, in the presence of multiple multipolar transitions channels, we found it more convenient to account for the effect of the proximity of nanoantenna via the modification of the field within the matrix elements appearing in the respective expressions. This allows us to account for potentially significant interference effects

between different transition mechanisms that contribute to the Hamiltonian (2).

Finally, we emphasize that the discussed scheme could be applied to an arbitrary emitter characterized by a discrete energy landscape and an arbitrarily shaped nanoantenna through the following steps:

1. The multipolar transition moments of the emitter $\mathbf{p}$, $\mathbf{m}$, and $\mathbf{Q}$ should be experimentally found or calculated (please see, e.g., the following data sources[25,26]).
2. The nanoantenna should be engineered such that in its optical response, expressed in terms of Purcell enhancement, desired multipolar resonances appear around the frequency of the pertinent quantum emitter's transition. This step might be used to modify the relative strength from different multipolar transition channels, in particular to balance two or more of them.
3. The field distributions around the emitter located in the nanoantenna surroundings should be evaluated and normalized according to the strengths of involved multipolar transition moments as follows from Eqs. (3)–(5). Please note that it is crucial to calculate all complex field components (electric, magnetic, and electric field derivatives) for each type of source, keeping their phase relations.
4. A coherent sum of multipolar contributions in Eq. (1) yields the total transition rate.

**Description of molecule**. An exemplary emitter that matches our requirements is the TDDFT-based model of an $OsO_3$ molecule, whose geometry is sketched in Fig. 1. The molecule is of the $D_{3h}$ symmetry that defines the selection rules for optically driven state transitions[27]. According to these selection rules, only certain matrix elements of multipolar moment operators, calculated for the states between which the transition occurs, can be nonzero[28,29]. We have obtained these values using the TDDFT method[30] (please see Methods; please also note a comment on robustness of these results with respect to applying different exchange-correlation functionals within DFT or small geometry changes).

The first (degenerate) pair of excited states obtained within TDDFT method is roughly 1.5 eV above the ground state, but the ED, MD, and EQ transition moments are zero. This is because

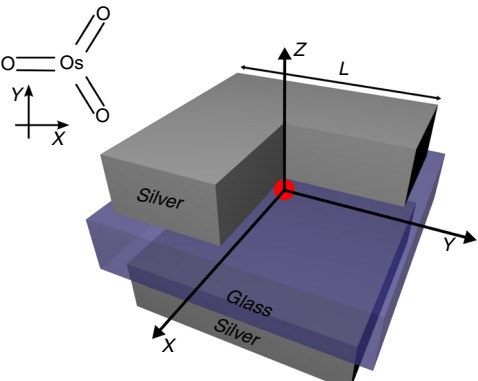

**Fig. 1 Geometries of nanoantenna and example molecule**. The nanoantenna is made of two cuboid silver patches of size $L \times L \times 50$ nm, separated with a 30-nm thick dielectric spacer with a side length $L + 20$ nm. Please note the upper cuboidal patch is identical to the lower one, while the cut in the figure serves only visualization purposes, so that the location of the molecule can be seen. Geometry of the $OsO_3$ molecule is sketched in the corner. The molecule is positioned in the center of the nanoantenna indicated by the red dot, with all atoms in the $xy$ plane.

these transitions result from spin-forbidden singlet-triplet excitations from the non-degenerate HOMO (highest-occupied molecular orbital) to the twofold degenerate LUMO (lowest unoccupied molecular orbital) of $e''$ symmetry. The next degenerate pair of electronic states of the molecule was found at 2.24 eV above the ground state, resulting from the corresponding (spin-allowed) singlet excitations. This corresponds to the free-space excitation wavelength of 553 nm, and is well isolated from the next electronic transition at 2.98 eV (416 nm). The two degenerate states at 553 nm, which we will denote with indices $j = 1, 2$, form a basis of a two-dimensional irreducible representation $E''$ of $D_{3h}$, while the ground state 0 is fully symmetric ($A'_1$). In theory, the allowed transitions between the ground and the excited states, i.e., those with nonzero transition matrix elements, are those coupled by the $m_x$ and $m_y$ components of the magnetic-dipole moment, and $Q_{yz}$ and $Q_{xz}$ components of the electric-quadrupolar moment[28,29]. Indeed, we find in the TDDFT simulations both transitions between the ground state 0 and the excited states $j = 1, 2$ introduced above, which we will refer to as "transition 1" or "transition 2", to be electric-dipole forbidden. The components of their electric-dipole moments $\mathbf{p}^{0j}$, that would enter the Hamiltonian in Eq. (2), are zeros.

The only nonzero components of the magnetic dipole $\mathbf{m}^{0j}$ and electric quadrupole $\mathbf{Q}^{0j}$ moments are $m_y^{01} = im$, and $Q_{xz}^{01} = Q$ for transition 1, and $m_x^{02} = im$ and $Q_{yz}^{02} = -Q$ for transition 2. Here, $i$ stands for the imaginary unit, $m = 0.420$ a.u. $\approx 7.79 \times 10^{-24}$ JT$^{-1}$ and $Q = 0.16$ a.u. $\approx 7.2 \times 10^{-41}$ Cm$^2$. According to Eq. (4), resulting free-space spontaneous emission rates read $\Gamma_0^{MD} \approx 112.6$ Hz and $\Gamma_0^{EQ} \approx 0.067$ Hz. This indicates that the quadrupolar channel is rather weak and the magnetic dipolar character dominates.

Below we describe a nanoantenna designed to match the character of this particular molecule: it sustains a resonant optical response to a magnetic dipolar source, and a significantly stronger one with respect to an electric-quadrupolar illumination. In this way the nanoantenna is aimed to restore balance between strengths of the two transition mechanisms.

Moreover, we will consider the molecule to be positioned in a dielectric matrix. Atomic-scaled inhomogeneities in the surroundings might in principle trigger breaking of selection rules, giving rise to nonzero, but small, electric-dipole moments. Furthermore, the fact (not taken into account here) that the symmetry of the molecule being in one of its emitting excited states may be lower than $D_{3h}$ due to the Jahn-Teller effect[31], would change the selection rules of the considered transitions. The inclusion of the spin-orbit interaction into the theoretical model would slightly modify the physical character of the excited electronic states of $OsO_3$ molecule and, as a

consequence, the selection rules and the transition moments. Finally, transitions from singlet states 1 and 2 to the lower pair of corresponding triplet excited states are not taken into account in our analysis. Radiative transitions between these states are spin-forbidden and thus probably not very relevant. However, intersystem crossing into the lower triplet states, i.e., a nonradiative relaxation, might quench the emission from the states under interest. For all these reasons it is important that the nanoantenna does not support electric-dipole sources, limiting their potential influence.

**Tuning nanoantenna for selected multipolar sources**. The nanoantenna that matches all above requirements consists of two silver patches and a dielectric spacer in between, similar to the one discussed in ref. [32]. A schematic of the structure is shown in Fig. 1. Such geometry could be fabricated using a pick-and-place technique in which an atomic-force-microscope tip is used to position nanoscaled objects at desired locations[33,34] or through Pattern Transfer Nano Manufacturing[35,36]. Other candidate geometries providing considerable magnetic field enhancement and electric field modulations are split ring resonators[9], diabolo antennas[37], metallic[14,15], or dielectric[12,38] dimers, etc.

Since the considered transition wavelength is in the optical regime, silver was chosen rather than gold due to strong absorption in gold at optical frequencies. Silver was modeled using the experimental data from ref. [39]. The size $L$ of the quadratic patches is varied in the simulations, while their thickness is set to 50 nm. To avoid unphysically sharp edges, the silver patches are modeled as rounded with a radius of curvature of 5 nm. The gap between the patches is 30 nm wide, and is filled with a symmetrically positioned quadratic dielectric spacer of length $L + 20$ nm. The permittivity of the dielectric is chosen as $\epsilon = 2.25$.

We now study the response of the described nanoantenna to ED, MD, and EQ sources of basic orientations with respect to the nanoantenna geometry (see Supplementary Note 1 and Supplementary Fig. 3). These basic orientations are parallel to the $x$ or $z$ axis of the coordinate frame from Fig. 1 (in case of dipoles or quadrupole with only diagonal elements), and in the $yz$ or $xy$ plane (in case of off-diagonal quadrupoles).

The response of the nanoantenna is studied in terms of the radiative decay rate enhancement $F_{rad}^{MO,\varphi}(\mathbf{r}_0, \omega) = P_{na,rad}^{MO,\varphi}(\mathbf{r}_0, \omega)/P_0^{MO}(\omega)$ where $P_{na,rad}^{MO,\varphi}(\mathbf{r}_0, \omega)$ is the time-averaged power radiated by a given multipolar emitter oscillating at frequency $\omega$ and located at position $\mathbf{r}_0$ near the nanoantenna, and $P_0^{MO}(\omega)$ is the time-averaged power radiated by the emitter in free space. In Fig. 2, a response to different multipolar sources positioned in the center of the nanoantenna is plotted in function

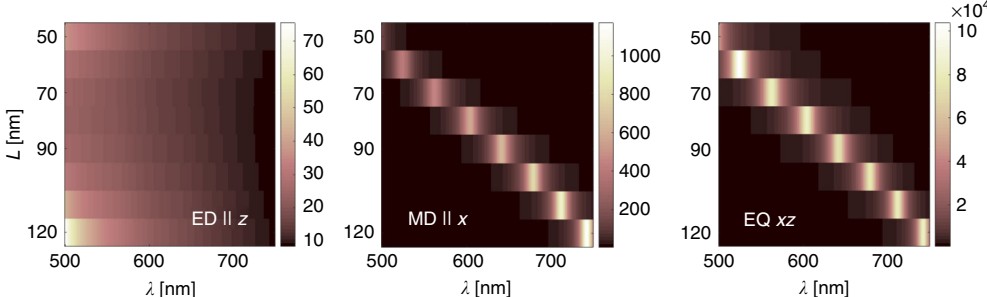

**Fig. 2 Total decay rate enhancement factors $F_{tot}$ for different optimally oriented multipolar sources**. The total rate includes radiative and absorptive contributions for an electric dipole parallel to the $z$ axis, a magnetic dipole parallel to the $x$ axis, and an off-diagonal electric quadrupole in the $xz$ plane. The enhancement is presented in function of free-space wavelength $\lambda$ of the source and for some discrete nanoantenna patch sizes $L$.

of free-space illumination wavelength $\lambda = 2\pi c/\omega$ and the size $L$ of the patches. Out of the basic orientations of each type of source, in Fig. 2 we only show results obtained for the one that provides the strongest response. Complete characteristics for all sources in different orientations can be found in Supplementary Note 3 and Supplementary Figs. 4 and 5.

The nanoantenna sustains a resonant response when illuminated with a magnetic dipole or an off-diagonal electric-quadrupole source, and the resonant wavelength is tunable with $L$. The enhancement factor is significant and reaches over two orders of magnitude in the MD, and over four orders of magnitude in the EQ case. Resonances due to these two types of sources spectrally overlap and the quadrupolar one dominates by two orders of magnitude—both properties suited to match the requirements of the $OsO_3$ transition indicated in the previous subsection. Contrary, the ED and the diagonal EQ sources do not show resonances in the considered parameter range. This is actually an advantage, since the possible influence of the ED channel that might be unlocked, e.g., due to local inhomogeneities in the molecular surroundings, is suppressed.

We proceed to analyze a response to a source that combines the MD and EQ components of orientations and strengths corresponding to the $OsO_3$ transitions indicated above.

**Interference of multiple transition channels**. Based on the results in Fig. 2, we now fix the nanoantenna patch size at $L = 75$

nm to localize its MD and EQ resonances around the $OsO_3$ transition wavelength of 553 nm.

Transition rates $\Gamma_{0j}$ for each transition $j = 1, 2$ were obtained for different orientations of a source, positioned in the center of the nanoantenna, in terms of Fermi's golden rule [Eq. (1)], with electric and magnetic fields calculated as described in Methods and Supplementary Note 2, and normalized according to the method described above in Different multipolar contributions to transition rates.

The interaction Hamiltonian in Eq. (2) consists of multiple contributions that can interfere. The relative impact of interference effects on the transition rate is expressed through the ratio

$$R = \frac{\Gamma - \left(\Gamma^{ED} + \Gamma^{MD} + \Gamma^{EQ}\right)}{\Gamma^{ED} + \Gamma^{MD} + \Gamma^{EQ}}. \tag{6}$$

A positive (negative) value of $R$ indicates a constructive (destructive) interference, $R = -1$ denotes a completely destructive process, $R = 0$ means interference effects are absent, and $R = 1$ is a fully constructive effect.

Figure 3 shows the transition rates (panels a & c) and $R$ (panels b & d) evaluated for a source, representing an $OsO_3$ molecule, positioned in the center of the nanoantenna, but rotated around axis $x$ (a & b) or $y$ (c & d). The initial orientation of the molecule (more precisely: transition elements of its multipolar moments $\mathbf{m}$ and $\mathbf{Q}$), shown in the inset of panel a, is such that the magnetic-dipole moment of transition 1 is parallel

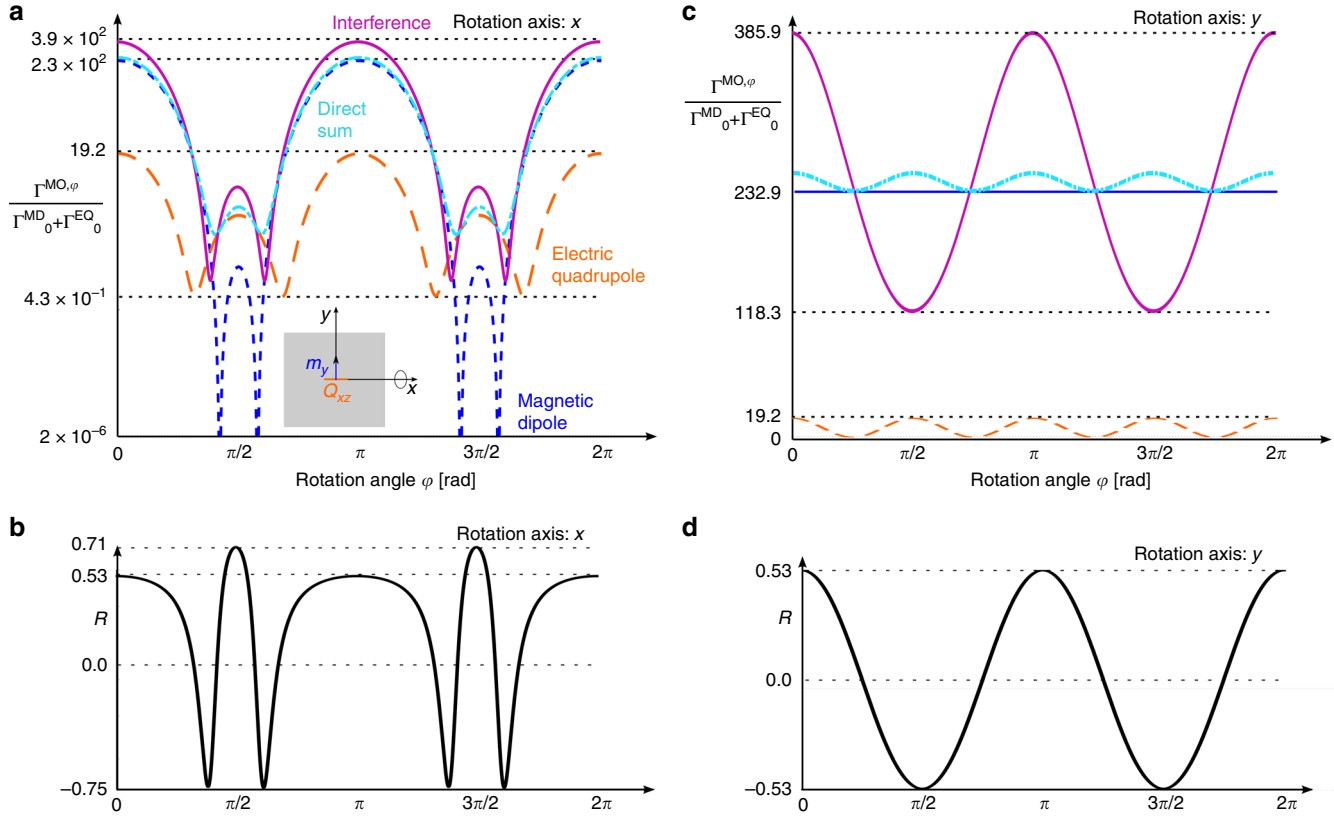

**Fig. 3 Calculated transition rates and *R* parameters**. Transition characteristics due to different multipolar mechanisms of a model $OsO_3$ molecule positioned in the center of the nanoantenna are shown in function of rotation angle with respect to $x$ (**a** and **b**) or $y$ axis (**c** and **d**). Original orientation of multipolar moments of transition 1 is depicted in the inset of panel (**a**). **a** Transition rates between the ground state 0 and excited state $j = 1$. The magnetic dipole $\Gamma^{MD,\varphi}$ and electric quadrupole $\Gamma^{EQ,\varphi}$ contributions are shown with dotted blue and dashed orange lines, respectively. The total transition rate $\Gamma$ (magenta solid line) includes the interference between the two contributing mechanisms, and may differ from their direct sum $\Gamma^{MD,\varphi} + \Gamma^{EQ,\varphi}$ (cyan dash-dotted line). **b** Figure of merit $R$ indicating the respective role of the interference terms for the full transition rate shown in (**a**). **c, d** As in panels (**a**), (**b**), but for rotations of the molecule around the $y$ axis of the coordinate frame.

to the $y$ axis of the coordinate system. This orientation is most favorable in the context of enhancement, as demonstrated for selected rotation axes in Fig. 3. In Fig. 3a we show the transition rates as functions of a rotation angle $\varphi \in (0, 2\pi)$ around the $x$ axis of the coordinates frame, normalized to the free-space transition rate $\Gamma_0^{MD} + \Gamma_0^{EQ}$. In panel a we have used logarithmic scale. The magnetic dipole $\Gamma_{01}^{MD,\varphi}$ (dotted blue lines) and electric quadrupole $\Gamma_{01}^{EQ,\varphi}$ (dashed orange) mechanisms contribute to the total transition rate $\Gamma_{01}$ (solid magenta). For a comparison, a direct incoherent sum $\Gamma_{01}^{MD,\varphi} + \Gamma_{01}^{EQ,\varphi}$ is shown in cyan dash-dotted line. We emphasize that the latter is not the complete characteristic of the transition, since it does not include interference effects.

Our first observation is that in the presence of the nanoantenna the transition rates through the initially weak channels, commonly considered as "forbidden", reach the regime of several tens of kHz. In free space, the transition rates for given moments $\mathbf{m}_{01}$ and $\mathbf{Q}_{01}$ are $\Gamma_0^{MD} \approx 112.6$ Hz and $\Gamma_0^{EQ} \approx 0.067$ Hz. With the discussed nanoantenna, they are enhanced to $\Gamma_{01}^{MD,\varphi=0} \approx 26.2$ kHz in the magnetic dipole and $\Gamma_{01}^{EQ,\varphi=0} \approx 2.17$ kHz in the electric-quadrupole channel. Since we are not able to rotate independently the magnetic dipole and electric-quadrupole moment of the transition, for all orientations the response is dominated by the magnetic channel, while the electric quadrupolar one contributes mainly through the interference term. For most orientations of the molecule rotated around the $x$ axis, the two channels constructively interfere leading to a further increased full transition rate of up to $\Gamma_{01}^{max} \approx 43.5$ kHz, with $R = 53.1\%$ enhancement due to interference (Fig. 3a, b). For the most favorable molecular orientations with the magnetic-dipole moment parallel to the nanoantenna patch edges, the total transition rate exceeds the free space one by 385.9 times. Strongest constructive interference reaches 70.6%, but happens at orientations for which the transition rate is small and the constructive effect is less influential. However, even more interesting is the perspective of destructive interference, leading to transition rate suppression. In particular, around $\varphi \approx (0.5 \pm 0.13)\pi$ rad and $(1.5 \pm 0.08)\pi$ rad the phase difference between the magnetic dipolar and the electric-quadrupolar response provides destructive interference with $R = -74.8\%$ and transition rate minima of $\Gamma_{01}^{min} \approx 73.23$ Hz, i.e., 64.5% of the free-space rate due to MD plus EQ mechanisms (Please note that in free space no interference between different multipolar mechanisms can occur, since a given type of emitter only gives rise to its characteristic field distribution decoupled from other multipolar sources at the same location.). Both the free-space rate and the nanoantenna-suppressed value are very small with respect to typical ED ones and correspond to large molecular lifetimes of the order of a hundredth of a second. The reason why the interference is not complete and transition is not fully suppressed is the non-optimal phase difference between the magnetic and electric fields at the molecular location, which could be improved through refined engineering. Please note that analysis of other sources of decoherence and dephasing, e.g., due to coupling to phononic bath, is beyond the scope of this work.

As the molecule rotates around the $y$ (Fig. 3c, d) and $z$ (not shown) axis of the coordinate frame, the MD mechanism dominates the transition and overcomes the EQ one by over 12 times for any rotation angle. Since in this case the rotation axis is parallel to the MD moment, a purely MD transition should not show any modulation. This is indeed confirmed by the blue dotted line in Fig. 3c. The influence of the weaker quadrupolar channel is, however, manifested through interference, which modifies the full transition rate by up to 53.1% (Fig. 3d) either in

the constructive or destructive way, depending on molecular orientation, leading to oscillations of significant amplitude.

Due to symmetry reasons, results for transition 2 can be obtained by a simultaneous interchange of transition indices $1 \leftrightarrow 2$ and rotation axes $y \leftrightarrow x$. If both states 1 and 2 were initially populated, the total emission rate would be an incoherent sum of the two corresponding contributions with weights determined by the population distribution. Since this would result in decreasing interference visibility, preparation of the molecule in only one of the states of the degenerate pair through selective pumping schemes would be beneficial.

## Discussion

The quantum emitter's orientation with respect to the nanoantenna plays a crucial role and should be under control in experiment (Fig. 3). Precise control of orientation of a single molecule over a wide range of angles is still challenging in nanoscaled systems. However, it is possible if a single quantum dot is exploited as an emitter instead, or if a coherent ensemble of emitters, including molecules, is used. In the case of nonpolar systems (i.e., without permanent dipole moments), the orientation of induced moments could be controlled with a laser beam[40], while polar systems could be relatively simply handled with electrostatic fields. Another possibility is to use defect ensembles in nanocrystals permanently oriented along a discrete set of directions related to the lattice structure. In this case, the group of defects oriented along a predefined direction can be selectively addressed with polarized light, as it was done in ref. [41].

In the example discussed above we have deliberately chosen a molecule with simple characteristics, with a transition involving only two multipolar channels to simplify our example and clearly discuss the role of interference. If the ED transition moment of the molecule would be nonzero, the electric-dipole term would need to be included in the analysis and might have a significant impact even though the discussed nanoantenna does not support ED resonances in the spectral region of interest. In general, an emitter combining all considered multipolar contributions could be studied, and interference effects are expected between any pair of multipolar transition mechanisms. Naturally, for a different emitter the nanoantenna design might need to be refined.

As the final remark we note that the investigated scenario could be understood in terms of the Kerker effect (for a review in the plasmonic context, please see ref. [42]). In the Kerker effect, a scattering particle is considered that supports at least two different multipolar contributions. Balancing them can be used to tailor the far-field scattering patterns upon illuminating the particle from an external source. In the standard realization with a combination of electric- and magnetic-dipole scatterer response, forward scattering is enhanced and backward scattering suppressed at the first Kerker condition due to interference of radiation from the two dipolar contributions. In the realization discussed in this work, the molecule plays the role of the source and the nanoantenna is the scatterer. Please note that the source is rather complicated with respect to the plane wave usually considered in a typical Kerker scenario. Also contrary to the standard realization, it is the near field where the crucial interference effects take place, namely in the volume surrounding the molecule. The scattered fields modulated with interference trigger a transition in the molecule at an enhanced or suppressed rate.

Combining results from quantum chemistry, nanooptics and quantum mechanics, we have established a general framework to investigate the interplay of different multipolar transition channels with plasmonic nanoantennas. We have shown tunability of the considered nanoantenna with respect to different multipolar contributions, that can be selectively enhanced through specific

illumination schemes. The nanoantenna was engineered to balance the rates corresponding to different multipolar transition mechanisms, unlocking the possibility of a considerable degree of interference. We have explored the possibility to superimpose and to coherently control the different emission pathways of quantum emitters originating from different multipolar contributions of the same transition. We have identified scenarios where the transition rate is enhanced but also suppressed below the free-space rate through the quantum-mechanical interference of the different transition pathways. In such a scenario, a specific emitter brought in a specific position and orientation relative to the nanoantenna may remain excited through lifetimes enhanced with respect to the transition lifetimes of an isolated molecule. Further exploiting the possibility of complete suppression of a certain transition through interference is the key for many applications in the context of quantum computing, quantum storage, and quantum communication.

## Methods

**Field characteristics**. For numerical simulations of optical response of the nanoantenna and field distributions, the commercially available software package CST Microwave Studio[43] has been used operating in the frequency domain. Details of implementation of different multipolar emitters can be found in the Supplementary Note 1 and Supplementary Figs. 1 and 2. Methods of calculation of fields and evaluation of the resulting transition rates can be found in the Supplementary Note 2.

**Molecular characteristics**. Molecular characteristics rely on the TDDFT implemented in the TURBOMOLE software[30,44–49]. The molecular structure of $OsO_3$ was optimized for the ground state with the functional BP86[50,51] using def-SV(P) basis sets[52] (BP86/def-SV(P)). Excitations were calculated at the BP86/def2-TZVPP level[53], and—for checking the influence of the functional—additionally with the hybrid functional B3-LYP[54]. For estimating the influence of spin-orbit coupling also the two-component variant of TDDFT[44,45,55] with respective bases, BP86/dhf-TZVP-2c[46] was used. The inclusion of spin-orbit coupling results in a small shift (by 0.02 eV) of the two excitation energies and in a negligible splitting (roughly by 0.001 eV). The changes in transition moments $\mathbf{m}^{0j}$ and $\mathbf{Q}^{0j}$ are small, around 2%. This is much less than changes upon using a hybrid functional such as B3-LYP instead of a pure one (ca. 0.1 eV for the energy and 10% for the transition moments). Also the influence of small changes in the Os–O distance (by 2 pm) on excitation energies, ca. 0.1 eV, is much larger than that of the spin-orbit coupling.

The output from the method includes a set of eigenstates characterized by transition energies from the ground state, as well as electric dipole, magnetic dipole, and electric-quadrupole transition moments. The multipolar transition moments are calculated according to[47,48]

$$p_k^{0j} = \sum_{i,a} \left(X^j + Y^j\right)_{i,a} \langle \varphi_i | \hat{p}_k | \varphi_a \rangle, \tag{7}$$

$$m_k^{0j} = \sum_{i,a} \left(X^j - Y^j\right)_{i,a} \langle \varphi_i | \hat{m}_k | \varphi_a \rangle, \tag{8}$$

$$Q_{kl}^{0j} = \sum_{i,a} \left(X^j + Y^j\right)_{i,a} \langle \varphi_i | \hat{Q}_{kl} | \varphi_a \rangle, \tag{9}$$

where $|\varphi_i\rangle$ ($|\varphi_a\rangle$) are occupied (virtual) Kohn–Sham orbitals, respectively[56], $X^j$ and $Y^j$ parametrize the transition density and can be calculated from the TDDFT response equation[49]. The multipolar moment operators in the SI system are defined as

$$\hat{\mathbf{p}}_k = -e\hat{r}_k, \tag{10}$$

$$\hat{\mathbf{m}}_k = \frac{ie\hbar}{2m_e}(\hat{\mathbf{r}} \times \nabla)_k, \tag{11}$$

$$\hat{\mathbf{Q}}_{kl} = -\frac{e}{2}\left(\hat{r}_k\hat{r}_l - \frac{1}{3}\delta_{kl}\hat{r}^2\right), \tag{12}$$

and the indices $k, l \in \{x, y, z\}$ enumerate components of vectors and tensors.

## Data availability
All relevant data are available from the corresponding author upon request.

## Code availability
Input files or sets of input parameters for TURBOMOLE, CST or MATLAB, as well as self-developed Python codes are available from the corresponding author upon request.

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

## Acknowledgements

The authors acknowledge Dr. Andrey Miroshnichenko for his advice on the implementation of the electric quadrupole and Alexander von Bernuth for his help with Fig. 1. C.R. and his team acknowledges support by the Deutsche Forschungsgemeinschaft (DFG, German Research Foundation)—project number 378579271—within project RO 3640/8-1 and from the VolkswagenStiftung. K.S. & P.G. acknowledge support from the Foundation for Polish Science (project no. Homing/2016-1/8) within the European Regional Development Fund and from the National Science Centre, Poland (project no. 2016/23/G/ST3/04045). J.S. acknowledges support from the Karlsruhe School of Optics and Photonics (KSOP). The authors also thank the Deutscher Akademischer Austauschdienst (PPP Poland) and the Ministry of Science and Higher Education in Poland. We acknowledge support by Deutsche Forschungsgemeinschaft, Open Access Publishing Fund of Karlsruhe Institute of Technology.

## Author contributions

C.R. and K.S. conceived the basic idea and managed the project. E.R. implemented the multipolar emitters. E.R., J.S., P.G., and M.G. made electromagnetic calculations. J.S. implemented the procedure to extract field around the source. M.K. and F.W. made TDDFT calculations and provided support on quantum-chemical methodologies. A.K. provided support on symmetry and molecular physics issues. K.S. implemented the code to evaluate transitions rates. K.S. and C.R. wrote the paper with the inputs from all authors.

## Competing interests

The authors declare no competing interests.
