## [Peer Review File · Nature Communications]

Reviewers' Comments:

Reviewer #1:

Remarks to the Author:

This manuscript reports a theoretical work on the use of metallic nanoantenna to tailor the interaction and interference between multipoles in emitting molecules. The general idea is to consider higher order multipole transitions in molecules and to design their optical environment in a way that enables tailoring the interference among these higher order transitions. By doing so, the authors show that it is theoretically possible to use these tailored interferences to either enhance or suppress the radiation of an emitting molecule at a specific wavelength by adjusting the antenna's geometry. The authors both used quantum-mechanical calculations and electromagnetic calculations to calculate the optical transitions in the OsO₃ molecule and to design an optical antenna that can tailor the interference between higher-order multipole transitions of this light emitter. This is an interesting and original idea. Researchers usually design antennas with specific multipolar resonances to tailor the scattered fields: here the authors consider the emitter itself as a multipolar antenna whose emission properties can be adjusted by making it interfering with itself through an external antenna.

However, I have a number of concerns that I have listed below:

1. The proposed system comprising the antenna and the emitter is specifically designed for the OsO₃ emitter. Indeed, based on symmetry considerations and quantum-chemical calculations, the authors have calculated the different multipole transitions allowed in this molecule. From these results, they design antennas that enhance both the magnetic dipole (MD) and electric quadrupole (EQ) transitions for this emitter to tailor the interferences among these two higher-order multipole transitions at the considered wavelengths. However, it is hard to envision how this method could be used in other systems and if it could be generalized to other emitting molecules. The OsO₃ is not a commonly used emitter so the authors should explain if this method could be generalized to other more "mainstream" emitters such as rare-earth ions or quantum dots.

2. As a related question, the interference between multipole transitions is designed for MD + EQ transitions. It is unclear if the same principle could be used to make other multipoles interfere, for example ED and MD transitions such as in ref. 18, or any other combinations?

3. It is shown that the interferences can be either constructive or destructive in function of the orientation of the molecule and the rotation angle. However, in a more realistic case, the system will be composed of an ensemble of isotropically oriented molecules. In this case, the overall optical response will be averaged over this ensemble of molecules. This is a very critical point because the authors' claim would therefore be no longer valid. Furthermore, there are no indications about how such a system of precisely oriented molecules could be fabricated in the future. The authors should comment on that and provide convincing arguments on how they envision a future realistic demonstration of their system.

4. The proposed antenna comprises from top to bottom a metallic patch, a dielectric spacer and another metallic patch. In this structure, the two metallic antennas have a rectangular shape with a side length L and the dielectric spacer in between has a side length $L+20\text{nm}$. This structure appears to me as not technologically feasible. Indeed, I don't see how a dielectric spacer sandwiched between two metallic nanostructures could have a side length longer than these two latter... The authors should convincingly explain how this device could be experimentally demonstrated and/or provide alternative geometries that could actually be fabricated (e.g. with a dielectric spacer the same length of the metallic patches).

5. One of the main outcome, as explained by the authors, is to suppress transitions through interferences. However, one usually wants to boost and enhance light emission. Could the authors explain in more details why it is useful to suppress optical transitions?

In conclusion, this manuscript starts on an interesting concept and shows some first proof-of-principle but I believe the different points I raised should be addressed before considering this paper for publication in Nature Communications.

Reviewer #2:

Remarks to the Author:

The manuscript presents a theoretical study of multipole sources enhancement with nanoantenna. The literature has considered up to now electric and magnetic dipolar sources and the novelty of the work reported is to consider multipolar sources and to show that constructive or destructive interferences may occur. The definition of enhancement factors for multipolar sources is derived in a straightforward and clear manner.

The authors illustrate the concepts by considering a specific case of OsO₃ molecules in a cavity made of two silver pads. The multipolar characteristics of OsO₃ are determined numerically thanks to TDDFT and finally the optical response is numerically obtained thanks to a commercial FDTD software. It is shown that due to interferences between the multipolar terms the transition rate could be enhanced or reduced.

Overall the manuscript brings a new concept in the field by considering a multipolar source and the effects that can be expected from interferences. I consider that it can be seen as a clever extension of the work that is done in the community and that the work is timely as there is a clear interest of several groups for this topic. However I also consider that it is not really a disruptive research but a clever incremental step.

Concerning the content of the paper itself, the conclusions are well supported by the data that are shown and the paper is well written thus I have no specific comment.

I suggest to add two references :

On multipole sources : Dodson, C. M., & Zia, R. (2012). Magnetic dipole and electric quadrupole transitions in the trivalent lanthanide series: Calculated emission rates and oscillator strengths. *Physical Review B*, 86(12), 125102.

On magnetic dipole sources : Rolly, B., Bebey, B., Bidault, S., Stout, B., & Bonod, N. (2012). Promoting magnetic dipolar transition in trivalent lanthanide ions with lossless Mie resonances. *Physical Review B*, 85(24), 245432.

Moreover I think interferences between multipole terms are already well known in a different context where the multipoles are related to the diffracted field by a particle. The Kerker effect for example is nothing more than interferences between electric and magnetic dipole terms and authors have already generalized the concept to multipolar terms. I think that the topic even if it is different is closely related to the work in this manuscript and I suggest to the authors to add a short discussion about this.

We would like to sincerely thank both referees very much for their thorough review, constructive feedback, and their support of our manuscript. In this document, we list all of the referee comments (shown in blue italic font), followed by our response (in black, regular font), with excerpts from the revised manuscript in green.

Reviewer #1:

1. The proposed system comprising the antenna and the emitter is specifically designed for the OsO₃ emitter. Indeed, based on symmetry considerations and quantum-chemical calculations, the authors have calculated the different multipole transitions allowed in this molecule. From these results, they design antennas that enhance both the magnetic dipole (MD) and electric quadrupole (EQ) transitions for this emitter to tailor the interferences among these two higher-order multipole transitions at the considered wavelengths. However, it is hard to envision how this method could be used in other systems and if it could be generalized to other emitting molecules. The OsO₃ is not a commonly used emitter so the authors should explain if this method could be generalized to other more “mainstream” emitters such as rare-earth ions or quantum dots.

As the referee has described, the antenna is engineered in accordance with the requirements imposed by a specific emitter, namely the OsO₃ molecule. In particular, magnetic dipole and electric quadrupole response of the antenna is tailored such that the corresponding resonances appear around a selected transition frequency of the molecule.

The same procedure could have been applied to any other emitter, including rare-earth ions and quantum dots, as we shortly discuss below. We emphasise that for the sake of demonstrating our approach we have to choose a specific example, but the work of course is not limited at all to those ingredients (antenna and molecule) that were considered. To make quantitative predictions about transition rates through different multipolar mechanisms of a selected emitter, the following steps should be taken:

1. The multipolar transition moments of the emitter should be known either from calculations or from experiment.
2. The antenna should be designed to support resonances around the frequency of the quantum emitter's transition under study.
3. The scattered fields should be evaluated according to the strengths of involved multipolar transition moments.

Please note that interference effects should only be expected if the same transition is characterized by at least two different multipolar moments. This is not unusual: in fact, magnetic dipole and electric quadrupole operators may share the same spatial symmetry and related transition moments are commonly nonzero simultaneously [see, e.g., *Electric-Quadrupole and Magnetic-Dipole Radiation in Linear Molecules. Applications to 1Π—3Π Transitions*, Ying-Nan Chiu, J. Chem. Phys. 42(8), 2671-2681 (1965) or *Magnetic dipole and electric quadrupole transitions in the trivalent lanthanide series: Calculated emission rates and oscillator strengths*, C. M. Dodson & R. Zia, Physical Review B, 86(12), 125102 (2012)].

In the discussed work, our goal was to consider an uncomplicated proof-of-principle example, with simple transition characteristics:

- We have decided on an emitter described by the D_{3h} group, whose symmetry imposes simple selection rules. In the particular case of transitions between states of A'_1 and E'' symmetries only

two out of three considered multipolar channels are allowed: the electric quadrupole and the magnetic dipole, while they are forbidden through the electric dipole channel.

- We have chosen a small molecule because electronic transitions of small systems are in general better spectrally isolated from each other. In the case of the O_3 molecule, the next pair of excited states with nonvanishing transition moments appears 3.44 eV above the ground state, which corresponds to free-space wavelength of 360 nm (the considered transition energy is 2.24 eV \sim 553 nm), as follows from our TDDFT calculations.

We emphasize that this choice is only dictated by the need for clarity of the example. In the case of emitters characterized by more complicated transition characteristics, identification of different competing effects would of course be possible on the theory level, but not necessary to convey the message of the manuscript.

Considering “more mainstream” emitters: Although the energetic landscape of rare-earth ions might be complicated, well isolated transitions are available and are typically chosen for applications. However, their reliable quantum-chemical characteristics require calculations with multi-reference methods, not capable within standard TDDFT framework. We have found the involved technical issues to be beyond the scope of our manuscript, but in practice the required transition moments can be calculated or measured. In fact, calculated transition moments for trivalent lanthanide ions can be found in a work suggested to be cited by Referee 2 [*Magnetic dipole and electric quadrupole transitions in the trivalent lanthanide series: Calculated emission rates and oscillator strengths*, C. M. Dodson & R. Zia, *Physical Review B*, 86(12), 125102 (2012)], and in references within.

The case of quantum dots is especially appealing, since with a proper choice of symmetry and geometry of the dot, polarizabilities of different multipolar orders can be tailored and large values can in principle be achieved. The ground and excited states might here correspond to the lack or presence of an exciton - this case would again be described with a relatively simple energetic structure. We consider the quantum-dot design to be an involved physical problem on its own and wish to omit this level of complexity in our manuscript.

To make the universal validity of our approach more clear, we have added the following statements summarizing the introduction of the method on page 8 of the revised manuscript:

Finally, we emphasize that the discussed scheme could be applied to an arbitrary emitter characterized by a discrete energy landscape and an arbitrarily shaped nanoantenna through the following steps:

1. The multipolar transition moments of the emitter \mathbf{p} , \mathbf{m} , and \mathbf{Q} should be experimentally found or calculated (please see, e.g., the following data sources [*Electric-Quadrupole and Magnetic-Dipole Radiation in Linear Molecules. Applications to $1\Pi-3\Pi$ Transitions*, Ying-Nan Chiu, *J. Chem. Phys.* 42(8), 2671-2681 (1965) or *Magnetic dipole and electric quadrupole transitions in the trivalent lanthanide series: Calculated emission rates and oscillator strengths*, C. M. Dodson & R. Zia, *Physical Review B*, 86(12), 125102 (2012)]).
2. The nanoantenna should be engineered such that in its optical response, expressed in terms of Purcell enhancement, desired multipolar resonances appear around the frequency of the pertinent quantum emitter's transition. This step might be used to enhance or suppress the relative strength from different multipolar transition channels, in particular to balance two or more of them.

3. The field distributions around the emitter located in the nanoantenna surroundings should be evaluated and normalized according to the strengths of involved multipolar transition moments as follows from Eqs. 3-5. Please note that it is crucial to calculate all complex field components (electric, magnetic, and electric field derivatives) for each type of source, keeping their phase relations.
4. A coherent sum of multipolar contributions in Eq. 1 yields the total transition rate.

2. As a related question, the interference between multipole transitions is designed for MD + EQ transitions. It is unclear if the same principle could be used to make other multipoles interfere, for example ED and MD transitions such as in ref. 18, or any other combinations?

Interference of the two dipoles, magnetic and electric, is in fact a very interesting physical question. It is of importance in particular in the context of optical activity, where quantum emitters are supposed to sustain both of these dipole moments simultaneously. We stress again that the method proposed in our manuscript can be used immediately to include electric dipole transitions in systems with a nonzero transition dipole moment. It's just that for the molecular transition discussed in the manuscript, the electric dipole moment vanishes: $p=0$. Such choice of emitter, as mentioned in the discussion of the previous point, is only dictated by simplicity reasons. However, there are many emitters for which ED and MD/EQ are simultaneously nonzero. According to the selection rules, these emitters should not have a centre of symmetry. Chiral systems supporting circular dichroism are an important, but not only example.

Regarding the nanoantenna design, one might want to spectrally overlap the two resonances, i.e. the electric and magnetic dipolar resonance, or, in contrast, to tailor the antenna such that it would mainly support the weaker channel to balance the usual free-space disproportion among the different channels. While both options are possible, the actual design for a specific case would depend on the selected emitter.

To entirely accommodate the question of the referee, we have added at the end of the *Results and Discussion* section of the revised manuscript the following discussion, which we hope clarifies this question:

In the example discussed above we have deliberately chosen a molecule with simple characteristics, with a transition involving only two multipolar channels to simplify our example and clearly discuss the role of interference. If the electric-dipole transition moment of the molecule would have been non-zero, the electric dipole term would have been required to be included in the analysis and might have had a significant impact even though the discussed nanoantenna does not support electric dipole resonances in the spectral region of interest. In general, an emitter combining all considered multipolar contributions could be studied, and interference effects are expected between any pair of multipolar transition mechanisms. Naturally, for a different emitter the nanoantenna design would have been required to be refined.

3. It is shown that the interferences can be either constructive or destructive in function of the orientation of the molecule and the rotation angle. However, in a more realistic case, the system will be composed of an ensemble of isotropically oriented molecules. In this case, the overall optical response will be averaged over this ensemble of molecules. This is a very critical point because the authors' claim would therefore be no longer valid. Furthermore, there are no indications about how such a system of precisely oriented molecules could be

fabricated in the future. The authors should comment on that and provide convincing arguments on how they envision a future realistic demonstration of their system.

The referee, of course, is correct on the one hand: In an ensemble of randomly oriented molecules the discussed effect would be averaged out. On the other hand, however, in recent years we have witnessed multiple experimental efforts geared towards developing our ability to access first of all but in perspective also to handle individual molecules in their position and orientation relative to optical antennas. Just one option would be to place molecules in a junction of a scanning tunneling microscope and study the light emission from this junction upon excitation with external light and not with electrical currents [e.g., *Single-molecule junctions beyond electronic transport*, Sriharsha V. Aradhya & Latha Venkataraman Nat. Nanotechnology 8, 399–410 (2013)]. Therefore, we deem the system considered in our manuscript to be of experimental relevance. While we describe selective effects to be observable, we are confident that it will motivate efforts to actually study them also in experiments. We elaborate in the following on a few approaches that would circumvent the problem of experimental accessibility:

- The first possibility is to indeed use a single quantum emitter per nanoantenna. In practise often a large number of nanoantennas is fabricated, and a diluted layer of quantum emitters is deposited on top. Then, the probability of finding a system (a nanoantenna + a single emitter) in which the emitter is suitably positioned with respect to the antenna is significant [for one of numerous realizations please see *Fluorescence enhancement in large-scale self-assembled gold nanoparticle double arrays*, M. Chekini et al. J. of Appl. Phys. 118(23), 233107 (2015)]. This proposition could be realized
 - with symmetric quantum dots in which direction of induced (i.e. transition) multipolar moments would depend on polarization of external fields used to excite an exciton, or asymmetric quantum dots for which dynamic control might be challenging but which could be fabricated multiple times in different orientations with respect to the neighbouring nanoantennas [for an example with a fixed orientation see *Unidirectional emission of a quantum dot coupled to a nanoantenna*, A. G. Curto et al. Science, 329(5994), 930-933 (2010)],
 - with molecules like organic dyes, which have been already used in single-molecule plasmonic experiments. There, molecules were located within a nanoscaled gap and with some control over molecular orientation achieved through encapsulation of dyes in larger molecules [*Single-molecule strong coupling at room temperature in plasmonic nanocavities*, R. Chikkaraddy et al., Nature 535 (7610), 127-130 (2016)].
- Another possibility is to use crystalline defects or lanthanide ions embedded in nanocrystals. There, multiple defects are oriented in one of a few predefined orientations corresponding to the crystal lattice structure (for example, there are 4 orientations in a tetrahedral diamond lattice). If a nanocrystal contains a number of defects, one could address a target fraction with suitably polarized fields [*Superradiant emission from colour centres in diamond* A. Angerer et al., Nature Physics 14(12), 1168 (2018)]. Admittedly, in this case, one would not have a full control over orientations of defects but could rather probe a discrete set of available orientations.
- Ensembles of co-aligned molecules could be used instead of randomly oriented ones. In the case of polar molecules with a permanent electric dipole (most asymmetric molecules), a DC electric field would exert a torque that would align the molecules. In the case of nonpolar molecules, a

control over orientations of all molecules simultaneously could be achieved using AC fields which would induce dipoles corresponding to electric-dipole-allowed transitions in these molecules. A resonant optical pulse would then create an ensemble of aligned, excited molecules [*Aligning symmetric and asymmetric top molecules via single photon excitation*, M. J. Weida & C. S. Parmenter, J. Chem. Phys. 107, 7138 (1997)] which could eventually spontaneously relax to the ground state *via* different multipolar mechanisms. A nonresonant pulse on the other hand, would align the molecules in their ground state [*Aligning molecules with intense nonresonant laser fields*, J. J. Larsen et al., J. Chem. Phys., 111(17), 7774-7781 (1999)]. For these and other experimental techniques of aligning molecules please see the review: *Colloquium: Aligning molecules with strong laser pulses*, H. Stapelfeldt and T. Seideman, Rev. Mod. Phys. 75, 543 (2003). In each case, a nanoantenna geometry should allow access to the molecules with a laser beam.

Please note that the possibility to align molecules with external laser beam suggests an appealing possibility to dynamically change the orientation of the emitter or ensemble of emitters, i.e. to enhance or suppress spontaneous emission on demand.

We have added the following discussion in the revised manuscript at the end of the *Results and Discussion* section:

As evident from Fig. 4, the quantum emitter's orientation with respect to the nanoantenna plays a crucial role and should be under control in experiment. Precise control of orientation of a single molecule over a wide-range of angles is still challenging in nanoscaled systems. However, it is possible if a single quantum dot is exploited as an emitter instead, or if a coherent ensemble of emitters, including molecules, is used. In the case of nonpolar systems (i.e. without permanent dipole moments), the orientation of induced moments could be controlled with a laser beam [*Colloquium: Aligning molecules with strong laser pulses*, H. Stapelfeldt and T. Seideman, Rev. Mod. Phys. 75, 543 (2003)], while polar systems could be relatively simply handled with electrostatic fields. Another possibility is to use defect ensembles in nanocrystals permanently oriented along a discrete set of directions related to the lattice structure. In this case, the group of defects oriented along a predefined direction can be selectively addressed with polarized light, as it was done in Ref. [*Superradiant emission from colour centres in diamond* A. Angerer et al., Nature Physics 14(12), 1168 (2018)].

4. The proposed antenna comprises from top to bottom a metallic patch, a dielectric spacer and another metallic patch. In this structure, the two metallic antennas have a rectangular shape with a side length L and the dielectric spacer in between has a side length $L+20\text{nm}$. This structure appears to me as not technologically feasible. Indeed, I don't see how a dielectric spacer sandwiched between two metallic nanostructures could have a side length longer than these two latter... The authors should convincingly explain how this device could be experimentally demonstrated and/or provide alternative geometries that could actually be fabricated (e.g. with a dielectric spacer the same length of the metallic patches).

As we said already in the original submission, our choice of the nanoantenna geometry was dictated by previous works where similar structures were shown to provide substantial magnetic Purcell enhancement factors of 3 orders of magnitude [*Controlling magnetic dipole transition with magnetic plasmonic structures*, T. Feng, Optics letters 36(12) 2369-2371, (2011)]. The larger dielectric spacer was a part of that geometry and we've considered it as well. There are two possible fabrication routes of such nanostructures:

- The first is a pick-and-place technique. There, nanometer- up to micrometer-scaled objects are picked up by an AFM tip, manipulated in space, and then placed at a predefined position, as described in [*Versatile force-feedback manipulator for nanotechnology applications*, M. Jobin, et al., Review of Scientific Instruments 76(5), 053701 (2005)]. In this method, positioning of individual objects is possible with subnanometer resolution [*Development of a compact nano manipulator based on an atomic force microscope: For monitoring using a scanning electron microscope or an inverted optical microscope*, F. Iwata et al., in 2012 International Conference on Manipulation, Manufacturing and Measurement on the Nanoscale (3M-NANO) (pp. 22-27). IEEE]. We deem fabricating individual particles and placing them with such technique to be feasible. Moreover, such technique also seems to allow the placement of the antenna relative to the position and orientation of the quantum emitter, as also discussed in the comment above. The pick-and-place technique could be applied multiple times to systematically change the orientation of the nanoantenna relative to the quantum emitter to study the described interference features in a systematic manner.
- The other method is based on a technique called Pattern Transfer Nano Manufacturing [for detailed description of commercial solution please see www.laserfocusworld.com/home/article/16556321/patterntransfer-nanomanufacturing-for-microoptics-and-spectroscopy and www.thorlabs.com/newgrouppage9.cfm?objectgroup_id=12964 , for related publications see *Triggered self-assembly of magnetic nanoparticles*, L. Ye et al., Sci. Rep. 6, 23145, 1-9 (2016), *Magnetic-Field-Directed Self-Assembly of Programmable Mesoscale Shapes*, L. Ye et al., Adv. Func. Mater. 26, 3983 (2016)]. Consecutive steps of the method are depicted in Fig. 1 below: A pattern for desired elements is fabricated under a PDMS (polydimethylsiloxane) layer (steps 1-4), then the particles are transferred to the PMGI (polymethylglutarimide, step 5) or another desired substrate and finally the PDMS layer is removed (steps 6-7). Repeating these steps, one can sequentially build the desired stack. Please note that further steps in this figure below (steps 8-12) are not of relevance here.

Fig. 1. Steps to manufacture exemplary optical elements using fused silica with the PTNM method. Source: www.laserfocusworld.com/home/article/16556321/patterntransfer-nanomanufacturing-for-microoptics-and-spectroscopy

However, besides the fabricability of the considered antennas, we would like to emphasize that the effects we describe would persist of course in other geometries whose fabrication might be less sophisticated and which provide considerable magnetic field enhancement and electric field modulations, for example split ring resonators [*Tailoring magnetic dipole emission with plasmonic split-ring resonators*, S. M. Hein, & H. Giessen, *Physical review letters* 111(2), 026803 (2013)], diablo antennas [*Strong modification of magnetic dipole emission through diablo nanoantennas*, M. Mivelle et al., *ACS Photonics* 2(8), 1071-1076 (2015)], or even dielectric nanostructures, although in the latter case the overall enhancement of higher-order-multipolar transitions might be weaker. Naturally, actual fabrication limitations should be carefully considered at the design stage for a follow-up experiment.

We have added the following statements at the beginning of subsection *Tuning nanoantenna for selected multipolar sources: Purcell enhancement* in the revised manuscript:

Such geometry could be fabricated using a pick-and-place technique in which an atomic-force-microscope tip is used to position nanoscaled objects at desired locations [*Versatile force-feedback manipulator for nanotechnology applications*, M. Jobin, et al., *Review of Scientific Instruments* 76(5), 053701 (2005), *Development of a compact nano manipulator based on an atomic force microscope: For monitoring using a scanning electron microscope or an inverted optical microscope*, F. Iwata et al., in 2012 International Conference on Manipulation, Manufacturing and Measurement on the Nanoscale (3M-NANO) (pp. 22-27). IEEE] or through Pattern Transfer Nano Manufacturing [*Triggered self-assembly of magnetic nanoparticles*, L. Ye et al., *Sci. Rep.* 6, 23145, 1-9 (2016), *Magnetic-Field-Directed Self-Assembly of Programmable Mesoscale Shapes*, L. Ye et al., *Adv. Func. Mater.* 26, 3983 (2016)]. Other candidate geometries which provide strong magnetic field enhancement and considerable electric field modulations are split ring resonators [*Tailoring magnetic dipole emission with plasmonic split-ring resonators*, S. M. Hein, & H. Giessen, *Physical review letters* 111(2), 026803 (2013)], diablo antennas [*Strong modification of magnetic dipole emission through diablo nanoantennas*, M. Mivelle et al., *ACS Photonics* 2(8), 1071-1076 (2015)], metallic [*Strong enhancement of forbidden atomic transitions using plasmonic nanostructures*, A. M. Kern & O. J. F. Martin, *Physical Review A* 85(2), 022501 (2012), *Controlling the dynamics of quantum mechanical systems sustaining dipole-forbidden transitions via optical nanoantennas*, R. Filter et al. *Physical Review B* 86(3), 035404 (2012)] or dielectric [*Promoting magnetic dipolar transition in trivalent lanthanide ions with lossless Mie resonances*, B. Rolly et al. *Physical Review B* 85(24), 245432 (2012), *Magnetic and electric hotspots with silicon nanodimers* R. M. Bakker et al., *Nano Letters* 15(3), 2137-2142 (2015)] dimers, etc.

5. One of the main outcome, as explained by the authors, is to suppress transitions through interferences. However, one usually wants to boost and enhance light emission. Could the authors explain in more details why it is useful to suppress optical transitions?

In the proposed scenario there are two mechanisms involved to modify the transition rates:

- Each of the considered transition channels is individually enhanced through the usual Purcell mechanism, as follows from the Purcell enhancement plots in Fig. 3 of the main manuscript and Figs. S4 & S5 of the Supporting Information file. This result already has interesting implications:

transition channels which are relatively weak in free space here become considerably strong, reaching tens of kHz in the case of our exemplary emitter. Similar observations were previously made for different nanoantenna geometries, and we cite many related works in our manuscript.

- On top of that, the Purcell-enhanced rates can be additionally modulated with the interference mechanism. Both enhancement and suppression of transition rates through interference are of course possible, depending on the relative phases of different field contributions, or - simply speaking - on the orientation of the emitter. Enhancement is interesting since it might lead to an additional increase of signal, i.e. it would support the effects which are typically of interest in the community - as the referee mentions, this is what “one usually wants”. To provide quantitative intuitions, for two equally strong transition channels being in phase one should expect enhancement to be doubled with respect to the case without interference. In our opinion, the possibility of suppression is much more interesting and we explain the reasons in the following paragraph.

In quantum information science, photons are a natural choice for information carriers, while quantum emitters are routinely proposed as and already exploited as quantum memories [*Storage and retrieval of single photons transmitted between remote quantum memories*, T. Chaneliere et al., *Nature* 438(7069), 833 (2005)]: systems in which the information is stored and processed. The process of spontaneous emission imposes an important limitation on the possible storage time and on the rate at which errors occur. A transition rate suppressed through interference corresponds to a longer spontaneous emission lifetime. Please note that we could achieve the fundamental limit of transition rate suppression: for properly engineered strengths and relative phases of different field components, the transition could in principle be attenuated completely, leading to, in principle, unlimited coherence times! The physical reason is that there is no photonic mode around the molecule into which the excitation could decay to. It should be noted, however, that other decoherence channels may be present in parallel to spontaneous emission, e.g. dephasing by phonons in quantum dots or collisions in ensembles of movable molecules. These channels will eventually limit coherence times. They could be suppressed at low temperatures, but further analysis of this issue is beyond the scope of our investigation.

To summarize, an attenuated transition rate would be important for the fields of quantum information and communication, in particular for information storage and processing in quantum memories, potentially leading to error suppression and significantly longer coherence times.

This motivation was indeed not explained enough in the original manuscript, where it was only briefly mentioned in the final conclusion. We have made the following changes:

- Modified the final sentence of the abstract: “Our work suggests that placing a suitably chosen quantum emitter at a well defined position and orientation relative to a nanoantenna can fully suppress the transition probability, waiving the limitation imposed on the emitter’s coherence time by spontaneous emission that is important in the field of quantum information science.”
- Added the following text to the introductory part on page 5 of the revised manuscript: “The possibility of suppression is especially appealing since it implies correspondingly longer spontaneous emission lifetimes. This largely increased lifetime of the emitter is crucial for applications such as quantum information processing and storage, where a short lifetime of the emitter is a serious constraint. This constraint might be relaxed in schemes based on our work.”

Reviewer #2:

1. I suggest to add two reference :

On multipole sources : Dodson, C. M., & Zia, R. (2012). Magnetic dipole and electric quadrupole transitions in the trivalent lanthanide series: Calculated emission rates and oscillator strengths. Physical Review B, 86(12), 125102.

On magnetic dipole sources : Rolly, B., Bebey, B., Bidault, S., Stout, B., & Bonod, N. (2012). Promoting magnetic dipolar transition in trivalent lanthanide ions with lossless Mie resonances. Physical Review B, 85(24), 245432.

The first of the suggested references is a valuable source of data on transition properties of lanthanide ions, potentially very useful for readers and for ourselves. We have added it along with another reference regarding linear molecules in the revised manuscript. In point 1 of the fragment listing subsequent steps of the proposed method added as the summary of the introductory section, we have written:

1. The multipolar transition moments of the emitter \mathbf{p} , \mathbf{m} , and \mathbf{Q} should be experimentally found or calculated (please see, e.g., the following data sources [*Electric-Quadrupole and Magnetic-Dipole Radiation in Linear Molecules. Applications to 1 Π –3 Π Transitions*, Ying-Nan Chiu, J. Chem. Phys. 42(8), 2671-2681 (1965) or *Magnetic dipole and electric quadrupole transitions in the trivalent lanthanide series: Calculated emission rates and oscillator strengths*, C. M. Dodson & R. Zia, Physical Review B, 86(12), 125102 (2012)]).

The other reference fits very well the introductory discussion, where a sentence was modified to include the article by Rolly *et al.*:

Until now, the enhancement of magnetic dipole emission was studied both theoretically and experimentally near metallic [*Strong enhancement of magnetic dipole emission in a multilevel electronic system* S. Karaveli & R. Zia, Optics letters 35(20), 3318-3320 (2010), *Tailoring magnetic dipole emission with plasmonic split-ring resonators* S. M. Hein & H. Giessen, Physical review letters, 111(2), 026803 (2013), *Mapping and quantifying electric and magnetic dipole luminescence at the nanoscale* L. Aigouy et al., Physical review letters 113(7), 076101 (2014)] or dielectric [*Quantifying the magnetic nature of light emission* T. H Taminiau et al., Nature Communications 3, 979 (2012), *Promoting magnetic dipolar transition in trivalent lanthanide ions with lossless Mie resonances*, B. Rolly et al., Physical Review B, 85(24), 245432 (2012).] nanostructures or with focused laser beams [*Excitation of Magnetic Dipole Transitions at Optical Frequencies*, M. Kasparczyk et al. Physical review letters, 114(16), 163903 (2015)].

Additionally, we have cited the same reference at the beginning of subsection *Tuning nanoantenna for selected multipolar sources: Purcell enhancement*:

[...] geometries which provide strong magnetic field enhancement and considerable electric field modulations are split ring resonators [*Tailoring magnetic dipole emission with plasmonic split-ring resonators*, S. M. Hein, & H. Giessen, Physical review letters 111(2), 026803 (2013)], diabolito antennas [*Strong modification of magnetic dipole emission through diabolito nanoantennas*, M. Mivelle et al., ACS Photonics 2(8), 1071-1076 (2015)], metallic [*Strong enhancement of forbidden atomic transitions using plasmonic nanostructures*, A. M. Kern & O. J. F. Martin, Physical Review A 85(2), 022501 (2012), *Controlling the dynamics of quantum mechanical systems sustaining dipole-forbidden transitions via optical nanoantennas*, R. Filter et al. Physical Review B 86(3), 035404 (2012)] or dielectric [*Promoting magnetic dipolar transition in trivalent lanthanide ions with lossless Mie resonances*, B. Rolly et al.

Physical Review B 85(24), 245432 (2012), *Magnetic and electric hotspots with silicon nanodimers* R. M. Bakker et al., Nano Letters 15(3), 2137-2142 (2015)] dimers, etc.

2. Moreover I think interferences between multipoles terms are already well known in a different context where the multipoles are related to the diffracted field by a particle. The Kerker effect for example is nothing more than interferences between electric and magnetic dipole terms and authors have already generalized the concept to multipolar terms. I think that the topic even if it is different is closely related to the work in this manuscript and I suggest to the authors to add a short discussion about this.

We would like to thank the referee for drawing our attention to the Kerker phenomenon, which is indeed related to the issues discussed in our manuscript. We have added the following discussion to its revised version:

[...] the investigated scenario could be understood in terms of the Kerker effect [for a review in the plasmonic context, please see *Generalized Kerker effects in nanophotonics and meta-optics*, W. Liu & Y. S. Kivshar, Optics express 26(10), 13085-13105 (2018)]. In the Kerker effect, a scattering particle is considered that supports at least two different multipolar contributions. Balancing the multipolar contribution can be used to tailor the far-field scattering patterns upon illuminating the particle from an external source. In the standard realization with a combination of electric- and magnetic-dipole scatterer response, forward scattering is enhanced and backward scattering suppressed at the first Kerker condition due to interference of radiation from the two dipolar contributions. In the realization discussed in this work, the molecule plays the role of the source and the nanoantenna is the scatterer. Please note that the source is rather complicated with respect to the plane wave usually considered in a typical Kerker scenario. Also contrary to the standard realization, it is the near field where the crucial interference effects take place, namely in the volume surrounding the molecule. The scattered fields modulated with interference trigger a transition in the molecule at an enhanced or suppressed rate.

Finally, we would like to thank once again to both referees for their helpful comments and overall support. We hope that we have managed to properly address all concerns and clarify ambiguities.

Reviewers' Comments:

Reviewer #1:

Remarks to the Author:

The authors have properly addressed all my comments and concerns and I now find the revised version of the manuscript suitable for publication in Nature Communications

The authors would like to sincerely thank both referees for reviewing our manuscript, their valuable comments and suggestions and the overall support.